# The History and Science of the Major Birch Pollen Allergen Bet v 1

**DOI:** 10.3390/biom13071151

**Published:** 2023-07-19

**Authors:** Heimo Breiteneder, Dietrich Kraft

**Affiliations:** Division of Medical Biotechnology, Department of Pathophysiology and Allergy Research, Center of Pathophysiology, Infectiology and Immunology, Medical University of Vienna, 1090 Vienna, Austria; dietrich.kraft@aon.at

**Keywords:** Bet v 1, major birch pollen allergen, molecular allergology, PR-10-like protein family, allergen ligands

## Abstract

The term allergy was coined in 1906 by the Austrian scientist and pediatrician Clemens Freiherr von Pirquet. In 1976, Dietrich Kraft became the head of the Allergy and Immunology Research Group at the Department of General and Experimental Pathology of the University of Vienna. In 1983, Kraft proposed to replace natural extracts used in allergy diagnostic tests and vaccines with recombinant allergen molecules and persuaded Michael Breitenbach to contribute his expertise in molecular cloning as one of the mentors of this project. Thus, the foundation for the Vienna School of Molecular Allergology was laid. With the recruitment of Heimo Breiteneder as a young molecular biology researcher, the work began in earnest, resulting in the publication of the cloning of the first plant allergen Bet v 1 in 1989. Bet v 1 has become the subject of a very large number of basic scientific as well as clinical studies. Bet v 1 is also the founding member of the large Bet v 1-like superfamily of proteins with members—based on the ancient conserved Bet v 1 fold—being present in all three domains of life, i.e., archaea, bacteria and eukaryotes. This suggests that the Bet v 1 fold most likely already existed in the last universal common ancestor. The biological function of this protein was probably related to lipid binding. However, during evolution, a functional diversity within the Bet v 1-like superfamily was established. The superfamily comprises 25 families, one of which is the Bet v 1 family, which in turn is composed of 11 subfamilies. One of these, the PR-10-like subfamily of proteins, contains almost all of the Bet v 1 homologous allergens from pollen and plant foods. Structural and functional comparisons of Bet v 1 and its non-allergenic homologs of the superfamily will pave the way for a deeper understanding of the allergic sensitization process.

## 1. Events That Led to the Discovery of the Major Birch Pollen Allergen Bet v 1

Clemens von Pirquet, a true legend of immunology [1], observed that antibodies were not only part of protective immune responses but could also cause diseases. He coined the word “allergy,” derived from ancient Greek (allos = other and ergon = work), to generally describe a change in the reactive capability of the immune system. He presented his findings in 1906 in the Münchener Medizinische Wochenschrift, thus introducing the term “allergy” to the medical terminology.

In 1976, Dietrich Kraft [2] became the head of the Allergy and Immunology Research Group at the Department of General and Experimental Pathology, then of the Medical Faculty of the University of Vienna, which would become the Medical University of Vienna in 2004. In autumn 1983, Kraft turned his full focus to allergic diseases. Working in an outpatient allergy clinic in Vienna together with his colleague Herwig Ebner, he came to the insight that allergy test solutions and hyposensitization solutions could only be standardized based on pure recombinant molecules. Kraft elaborated on this idea during a lengthy stay in hospital due to a severe bicycle accident. Consequently, on 12 December 1983, he called Michael Breitenbach on the phone, as a meeting in person was impossible due to the unusually heavy snowfall that day. During this one hour call, Kraft managed to persuade Michael Breitenbach to contribute his essential expertise in molecular cloning to this newly hatched project. Thus, the keystone for the founding of the Vienna School of Molecular Allergology was put into place. Kraft also recruited Otto Scheiner and Helmut Rumpold, who brought additional expertise, i.e., immunohistochemistry and medical laboratory techniques, to the project.

From 1983 to 1984, investing in the cloning and production of recombinant allergens was of no interest for funding agencies in Austria. In a similar manner, pharmaceutical companies active in the allergy field in countries including Denmark, Sweden, Germany, and the USA did not support Kraft’s plans, deeming them to be too exotic. During a banquet held by the Austrian Society of Allergology and Immunology (ÖGAI), Kraft ended up sitting next to Jörg Mayrhofer, to whom he laid open his vision for recombinant allergens. Mayrhofer, then owner of the Schutzengel Apotheke (Guardian Angel Pharmacy) in Linz, Austria, showed an interest in Kraft’s plans as a potential business opportunity. During a follow-up visit to Linz, Jörg Mayrhofer and his father, Theodor Mayrhofer, agreed to fund Kraft’s ambitious plans. Consequently, in 1984, the company Biomay was founded that would finance all the formative years of Kraft’s team of molecular allergology.

As around 5% of the Austrian population suffer from pollinosis induced by birch pollen in early spring, and as there is only one major allergen present in birch pollen extract, this allergen was chosen as the target. Following the rules of allergen nomenclature laid down by the WHO/IUIS Allergen Nomenclature Sub-Committee [3] and the then-valid taxonomical name for the white birch, *Betula verrucosa*, this allergen was designated as Bet v 1. Michael Breitenbach selected Heimo Breiteneder as the person who would be responsible for the cloning of the cDNA for Bet v 1. Breiteneder had just finished his thesis on the genome organization of the cyanelles of *Cyanophora paradoxa* [4] under the guidance of Wolfgang Löffelhardt at the Institute of General Biochemistry of the University of Vienna.

On 2 May 1985, the work on the cloning of the Bet v 1 cDNA started in earnest. At first, the isolation of RNA from birch tissues needed to be established. This was done from roots of birch seedlings, and from leaves and inflorescences of mature trees. Finally, pure pollen from birch trees was obtained, RNA was isolated and poly(A)^+^ mRNA was enriched. This mRNA preparation was translated in vitro in a cell-free wheat germ system, and the proteins synthesized were separated by SDS-PAGE and transferred to nitrocellulose. When the blots were incubated with sera from patients allergic to birch pollen, IgE binding to a 17-kD protein, presumably Bet v 1, was observed, indicating the presence of allergen-coding mRNA in the RNA preparations [5]. Based on this finding, the *E. coli* phage λgt11 was selected to produce a cDNA expression library from birch pollen poly(A)^+^ mRNA. Michael Breitenbach was also instrumental in obtaining further knowhow from Arnold Bito and Klaus Richter at the Academy of Sciences in Salzburg on working with phage λgt11. The expression library was going to be screened, in a manner identical to immunoblots—with sera from birch pollen allergic patients—a combination of techniques that had not been previously used.

Phage λgt11 clones containing Bet v 1-encoding cDNAs, and thus producing recombinant Bet v 1 in infected *E. coli* cells, were first identified on 3 July 1988. The full cDNA sequence of Bet v 1 was published in the EMBO Journal in 1989 [6], representing the most abundant isoform in birch pollen, called Bet v 1.0101. Thus, Bet v 1 became the first cloned plant allergen and also the first allergenic PR-10-like protein that was published worldwide. Its sequence was similar to the N-terminal peptide sequences of the pollen allergens of hazel, alder and hornbeam, trees belonging to the order of Fagales and, therefore, closely related to birch [7]. What came as a surprise was that Bet v 1 also displayed a 55% sequence identity with a pea disease resistance response gene. Pea and birch being taxonomically quite distant, this was the first indication of a gene family whose founding member must have existed a long time ago. In 1991, the cDNA sequence and recombinant protein of the alder pollen allergen Aln g 1 was published [8], followed in 1993 by the major allergen of hazel pollen, Cor a 1 [9]. This publication describes the identification of four cDNA clones whose open reading frames coded for different isoforms of the major hazel pollen allergen. Interestingly, recombinant proteins of these four isoforms displayed differing IgE binding capacities. In 1995, the cDNAs and recombinant proteins of the major allergens of apple, Mal d 1 [10], and of celery, Api g 1 [11], joined the ranks of allergenic PR-10-like proteins, thus firmly establishing the presence of members of this gene family in the plant kingdom.

## 2. The PR-10-like Family of Allergenic Proteins

In 1980, van Lonn and colleagues defined pathogenesis-related proteins (PR proteins) as “proteins encoded by the host plant but induced only in pathological or related situations” and suggested the first groupings of such PR proteins into families [12]. Currently, 17 such families are described whose members are key components of the plant innate immune system [13]. Plants use PR proteins in inducible defense responses to combat various biotic and abiotic stresses. In contrast, Bet v 1, which is related by sequence to the PR-10 family of proteins, is constitutively expressed in pollen at rather high concentrations. Hence, the correct term for Bet v 1 and its homologous proteins is PR-10-like proteins. The Bet v 1 homologs are slightly acidic small (154–163 amino acid residues) and predominantly cytoplasmic proteins with molecular masses of around 17 kDa. In general, these proteins are susceptible to gastric digestion [14] and heat processing results in a loss of the native protein fold via denaturation, oligomerization and precipitation along with a subsequent reduction in IgE recognition [15]. Some exceptions exist; for example, the allergenicity of the Bet v 1 homolog from carrot, Dau c 1, is not destroyed by cooking [16].

Birch flowers in spring, and its pollen is one of the most common causes of IgE-mediated allergies in Northern and Central Europe as well as in North America [17]. The major sensitizing allergen present in birch pollen is Bet v 1 which is regarded as a marker allergen for a primary sensitization to pollen of birch and other Fagales trees (e.g., alder, hazel, hornbeam, beech, oak). However, allergic reactions to Fagales pollen can be initiated independently by PR-10-like proteins from pollen of all members of the Betulaceae and Fagaceae families (Table 1). However, 25% of the IgE-binding epitopes of PR-10-like allergens from the pollen of trees of the Betuloideae and Coryloideae families are unique to these subfamilies, while pollen allergens from the Fagaceae are generally cross-reactive [18].

Bet v 1 and its homologs in pollen are considered important inducers of birch pollen-associated plant food allergies [17]. The most frequently observed clinical entity, the oral allergy syndrome (OAS) [24], is caused by IgE antibodies that cross-react between Bet v 1 and its homologs in fruits, nuts, seeds and vegetables. Homologs of Bet v 1 have been identified in a wide range of plant foods (Table 2). Hence, Bet v 1-allergic patients are at risk to even react to novel foods without prior exposure.

In contrast to Bet v 1, Bet v 1-related food allergens are in general unable to act as primary sensitizers of predisposed individuals. The major reason for the very low ability of most plant food Bet v 1 homologs to induce sensitization is their high susceptibility to gastric digestion [14]. In addition to the not life-threatening symptoms of the OAS (e.g., itching, redness and tearing of the eyes, itch in the nose and oropharynx, sneezing, runny or stuffy nose, wheezing), severe reactions to Gly m 4, the Bet v 1 homolog from soybean, have been observed in a subpopulation of Bet v 1-allergic individuals [24,43].

## 3. Natural Ligands of PR-10-like Allergenic Proteins and Their Biological Functions

The three-dimensional structure of the major birch pollen allergen Bet v 1 was the first experimentally determined structure of a clinically important major inhalant allergen [44]. The so-called Bet v 1 fold (Figure 1) consists of a seven-stranded anti-parallel beta-sheet that wraps around a 25 residue-long C-terminal alpha-helix. The beta-sheet and the C-terminal part of the long a-helix are separated be two consecutive alpha-helices. The main structural feature of the Bet v 1 fold is a long forked cavity that penetrates the whole protein and that is solvent accessible via three openings to the protein’s surface. The volume of the cavity is about 1500 Å^3^ and its surface is predominantly hydrophobic. This hydrophobic cavity has the ability to bind a variety of experimental and physiologic ligands [45,46]. A detailed description of ligand classes interacting with PR-10 allergens is provided in the 2020 review by Aglas and colleagues [47]. McBride and colleagues chose various polyphenols, fatty acids, phenols, plant hormones, and one plant and one animal sterol to routinely screen for ligand binding characteristics of PR-10-like allergens [48]. However, studies on the physiologic natural ligands of PR-10-like allergens are still limited.

The first physiologic ligand of a PR-10-like protein was determined for Bet v 1. Quercetin-3-*O*-sophoroside (Q3OS), a glycosylated flavonol, was found as a natural ligand bound to Bet v 1 isolated from mature birch pollen [49]. Flavonoids contribute to pigment formation in flowering plants, are involved in plant hormone signaling, facilitate pollen tube formation and protect pollen DNA from UV radiation [50,51]. Flavonoids are stored in pollen as glycosylated precursors and, during the rehydration of the pollen grain, are processed into their active form by pollen glycosyltransferases. At this time point, Q3OS would need to be displaced from the Bet v 1-Q3OS complex to become accessible for deglycosylation. In solution, Bet v 1.0101 is conformationally heterogeneous and cannot be represented by a single structure [52]. NMR relaxation data suggest that structural dynamics are fundamental for ligand access to the protein interior. Complex formation then leads to a significant rigidification of the protein, along with a compaction of its three-dimensional structure, which is observed in the crystalized protein. Ligand binding stabilizes the conformation of Bet v 1, resulting in an increased melting point as well as drastically increased resistance towards endo-/lysosomal proteolysis [53]. The increased stability could hinder optimal proteolytic processing of the allergen, which would favor the development of a Th2 immune response. Analyses of human IgE binding on Bet v 1 in mediator release assays revealed that ligand-bound allergen-induced degranulation at lower concentrations. However, in basophil activation tests with human basophils, ligand-binding did not show this effect [53]. Phytoprostane E1 (PPE1) was identified as another physiologic ligand of Bet v 1 [54]. PPE1 interacts with Bet v 1, increasing its stability to proteolytic degradation. Pollen-derived PPE1 interacts with Bet v 1 with high affinity, increasing its stability and attenuating its degradation by processing by lysosomal cathepsin S. PPE1 also inhibited lysosomal cathepsins by blocking their catalytic cysteines. In addition, processing of Bet v 1.0101 and a hypoallergenic isoform differed distinctly, resulting in low- or high-density class II MHC loading and subsequently in Th2 and Th1 polarization, respectively [55]. Binding of glycosylated flavonoids to Bet v 1 isoforms is governed by the sugar moiety, and various isoforms show individual and highly specific binding behaviors for the different ligands [56]. It is tempting to speculate that the binding of phytoprostanes is also isoform-specific, resulting in the observed differences in Bet v 1 isoform allergenicity.

Natural Cor a 1 was extracted from mature hazel pollen, and its ligand was identified as quercetin-3-O-(2-O-β-D-glucopyranosyl)-β-D-galactopyranoside (Q3O-(Glc)-Gal) [57]. Similarly to nBet v 1, nCor a 1 is composed of different isoallergens and variants. The authors of this study were able to confirm the presence of the variants Cor a 1.0103 and Cor a 1.0104 on the basis of variant-specific tryptic peptides. Furthermore, four known Cor a 1 variants previously detected in hazel pollen and one isoform, Cor a 1.0401, detected only in hazel nuts [30] were analyzed as recombinant proteins for binding the ligand Q3O-(Glc)-Gal. Interestingly, Q3O-(Glc)-Gal was identified as a natural ligand of Cor a 1.0401 [57]. It was further demonstrated that Cor a 1.0401 and Bet v 1.0101 exhibited highly selective binding for their specific ligand but not for the respective ligand of the other allergen. A hypothesis of the role of Q3O-(Glc)-Gal in the unusual reproductive biology of hazel is presented by the authors of the study. The ligand is released from pollen, bound by Cor a 1.0401 present in the stigma, and hence in the nut, to prevent premature deglycosylation, and then only released after several months to be converted into quercetin to assist the formation of the secondary pollen tube.

Emanuelsson and co-workers reported that fruits of colorless white strawberry cultivars showed very low levels of Fra a 1 expression in contrast to red-colored fruits, as well as a downregulation of several enzymes in the pathway for the biosynthesis of flavonoids to which the red color pelargonidin belongs [58]. Casañal and coworkers presented crystallographic structures of Fra a 1 isoforms in complex with the naturally occurring flavonoid catechin, providing for the first time, a molecular basis for the function of these proteins in flavonoid biosynthesis [59]. The authors conclude that Fra a 1 proteins could act as transporters or “chemical chaperones” binding to flavonoid intermediates and making them available to processing enzymes, or they function as cytosolic transporters of flavonoids from the endoplasmic reticulum to other cellular membranes.

## 4. The Bet v 1-like Superfamily of Proteins

The determination of the Bet v 1 structure [44] paved the way for the search and definition of the Bet v 1-like superfamily. Radauer and colleagues used psc++, an improved version of the ProSup structural alignment program [60] to search the Protein Data Bank for structural homologs of Bet v 1 [61]. The resulting structures were then classified into eleven families, of which the Bet v 1 family was one. Today the so called Bet_v_1_like set, which corresponds to the original Pfam Clan CL0209, encompasses 25 families (https://www.ebi.ac.uk/interpro/set/pfam/CL0209/; accessed on 19 June 2023). The underscores in the allergen designation were added by the InterPro database. This designation is not in accordance with the official designation that was assigned by the WHO/IUIS Allergen Nomenclature Sub-Committee (http://allergen.org/viewallergen.php?aid=129, accessed on 19 June 2023). The most widely distributed families of the Bet v 1 superfamily were the polyketide cyclase family and the AHA1 (activator of Hsp90 ATPase homolog 1) family. Members of both families are found in bacteria, archaea and eukaryotes. The ring hydroxylases and the CoxG (named after the *cox g* gene encoding a subunit of the carbon monoxide dehydrogenase) families are widely distributed in bacteria and archaea and the StART (steroidogenic acute regulatory protein-related lipid transfer) family in bacteria and eukaryotes. The phosphatidylinositol transfer proteins are found only in eukaryotes, and members of the Bet v 1 family are exclusively present in plants.

The Bet v 1 family was classified into 11 subfamilies, including 9 from plants and 2 from bacteria [61]. The largest subfamily is the dicot PR-10 subfamily, which contains genuine PR proteins whose expression is upregulated upon pathogen infection, wounding or by abiotic stress, and PR-10-like proteins whose expression is developmentally regulated. The vast majority of allergens are found in the PR-10 subfamily. Allergenic members of other subfamilies were only rarely described and include the mung bean allergen Vig r 6 [34], a member of the subfamily of the cytokinin-specific binding proteins, and the kiwi allergen Act d 11 [26], a member of the major latex protein/ripening-related protein subfamily. So far, this limits the allergens to 3 of 11 subfamilies and to 1 of 25 families of the Bet v 1-like superfamily. The reason that allergenic members of the Bet v 1-like superfamily were found almost exclusively in the PR-10 subfamily is probably linked to the specific ligands they harbor (see Section 3 above), to other biologically active matrix components of the specific plant tissues and the ease or absence of exposure.

A truly ancient protein possessing the Bet v 1 fold must have existed as the last universal common ancestor of this superfamily, giving rise to proteins with diverse functions by insertion of additional structural elements or by fusion to other domains. During evolution, sequence similarities decreased to very low values, making sequence-based searches for Bet v 1 homologs unfeasible. The APE2225 (PDB 2NS9), a CoxG family member from the archaeon *Aeropyrum pernix* K1, has a fold identical to Bet v 1 (Figure 1) but only 14% sequence identity. *A. pernix* K1 is an aerobic hyperthermophilic archaeon isolated from a coastal hydrothermal vent at Kodakara-Jima Island, Japan [62]. This archaeon grows optimally at 90 to 95 °C, pH 7.0, and a salinity of 3.5%, which indicates a high stability of its Bet v 1 fold, a feature that was lost once the Bet v 1 predecessor gene moved into land plants. Recently, a bacterial Bet v 1-related protein, possibly a member of the polyketide cyclase family, TTHA0849 (Figure 1) [63] from *Thermus thermophilus* served as a non-allergenic scaffold to create chimeric proteins by grafting individual epitope-sized, contiguous surface patches of the allergen Bet v 1 onto its surface [64]. These chimeras were then used to determine patient-specific patterns of epitope recognition by IgE antibodies. The norcoclaurine synthases (NCS) form another subfamily of the Bet v 1 family [61]. The NCS from the plant meadow rue (*Thalictrum flavum*), a structural homolog of Bet v 1, does not bind Bet v 1-reactive IgE and was therefore used as another scaffold for grafting of a Bet v 1-specific IgE epitope [65].

## 5. The Yeast Connection

Yeast represents a highly useful genetic model system due to its easily manipulated genome and easily manageable methods for the functional analysis of gene products. Thus, yeast is easily suited to identify biological functions of new and unknown eukaryotic protein-coding genes. We have therefore searched for Bet v 1 homologs in the yeast genome and found a family of sterol transfer proteins recently described in the literature.

Sterols play a key role in regulating the fluidity and barrier function of plasma membranes in eukaryotic cells [66]. Sterol transport proteins distribute sterols from their point of synthesis in the endoplasmic reticulum (ER) to their exact subcellular localization. The ability to solubilize lipids into aqueous solutions is a general property of steroidogenic acute regulatory protein-related lipid transfer (StART) -like domains. Gatta and colleagues have identified a family of StART domain containing proteins that are anchored at membrane contact sites and that transport sterols from the ER to the plasma membrane [67]. The yeast *Saccharomyces cerevisiae* possesses six such proteins with StARkin domains [67,68,69] of which four (Lam1–Lam4) are anchored to the ER membrane. The StARkin domain (the name is derived from kin of steroidogenic acute regulatory protein) is an α/β helix-grip-fold structure with a deep hydrophobic pocket [70]. The major birch pollen allergen Bet v 1 represents the first described StARkin domain [44]. StARkin domains were used to define a large superfamily of proteins, which is also referred to as the Bet v 1-like superfamily [61] or START domain superfamily [71]. The name StARkin was proposed as a more inclusive name for this superfamily [72].

The name Lam is derived from “lipid transport proteins anchored at membrane contact sites”. They are integral membrane proteins anchored into the ER membrane by a C-terminal transmembrane helix [72]. The yeast proteins Lam1 and Lam3 have a single StARkin domain, whereas Lam2 and Lam4 each have two StARkin domains. Jentsch and colleagues carried out a structure-function analysis of the second StARkin domain of Lam4 (Lam4 SD2), showing that Lam4 SD2 undergoes conformational changes upon binding sterol and that it catalyzes sterol transport between vesicles in vitro [73]. Crystal structures of Lam4 SD2 with and without bound 25-hydroxycholesterol were also obtained. Tong and colleagues obtained crystal structures of both StARkin domains of Lam2 and Lam4 in the apo form and of Lam2 SD2 in complex with ergosterol [74].

To identify proteins with similar structures to Bet v 1.0101, we used the online available Vector Alignment Search Tool+ (https://structure.ncbi.nlm.nih.gov/Structure/VAST/vast.shtml, accessed on 13 July 2023) [75,76]. As a bait, NMR data (PDB: 6R3C) of the allergen were chosen [57]. Figure 2 shows ribbon representations of Bet v 1.0101 and the StARkin domains of Lam2 and Lam4, whereby Lam2 SD2 is only available as a swapped dimer. Figure 3 shows an overlay of Bet v 1.0101 and the first StARkin domain of Lam4 (PDB 5YQJ) [74], one of the results of the VAST+ analysis.

## 6. Conclusions

The research on Bet v 1 has spanned five decades. It started in the 1980s with an allergy-driven focus and has expanded into numerous and highly diversified studies of the functions of the members of the Bet v 1-like/StARkin superfamily of proteins. The first described StARkin domain was Bet v 1. Identification of ligands of Bet v 1 and its homologs offer the first insights into the varied functions of these proteins. However, information on ligands and functions of the various isoforms of Bet v 1 in birch pollen is still missing, as it is unclear why there are so many isoforms present. The Bet v 1 scaffold is a very old and versatile one and has been used for a wide variety of functions. We do not believe that the Bet v 1 homologs present in birch or in other organisms are the result of convergent evolution. Rather, it is a very useful scaffold that was conserved and used in many different ways. The Bet v 1-like/StARkin superfamily keeps growing, and new members are still being discovered.

## Figures and Tables

**Figure 1 biomolecules-13-01151-f001:**
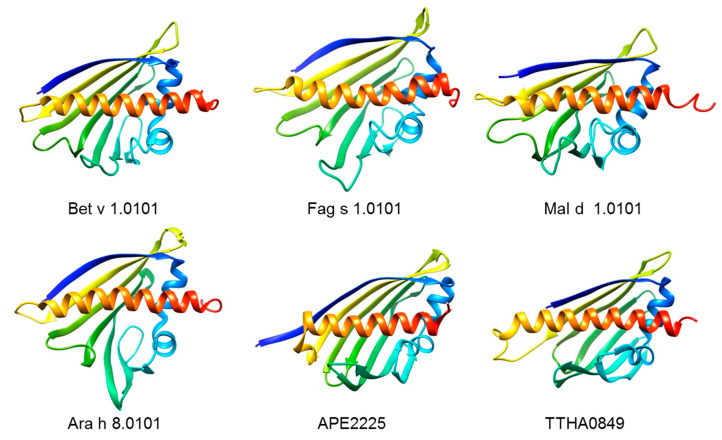
Ribbon representations of birch pollen Bet v 1 (PDB 4A88) and homologs from beech pollen (Fag s 1; PDB 6ALK), apple (Mal d 1; PDB 5MMU), peanut (Ara h 8; PDB 4M9B), the hyperthermophilic archaeon *Aeropyrum pernix* (APE2225; 2NS9) and the extremely thermophilic bacterium *Thermus thermophilus* (TTHA0849; PDB 2D4R) rainbow-colored from blue at the N-terminus to red at the C-terminus. The 3D images were created with the molecular modeling system UCSF ChimeraX (https://www.rbvi.ucsf.edu/chimerax/, accessed on 9 June 2023).

**Figure 2 biomolecules-13-01151-f002:**
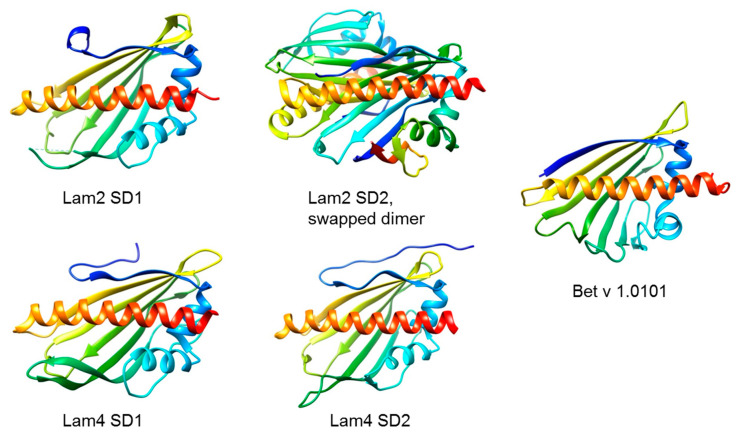
Ribbon representations of Bet v 1 (PDB 4A88) homologous structures from the yeast *Saccharomyces cerevisiae*. The StARkin domains of Lam2 (StARkin domain 1, PDB 5YQI; StARkin domain 2 as a swapped dimer, PDB 5YQQ) and of Lam4 (StARkin domain 1, PDB 5YQJ; StARkin domain 2, PDB 5YQP) are shown. The structures are rainbow-colored from blue at the N-terminus to red at the C-terminus. The 3D images were created with the molecular modeling system UCSF ChimeraX (https://www.rbvi.ucsf.edu/chimerax/, accessed on 19 June 2023).

**Figure 3 biomolecules-13-01151-f003:**
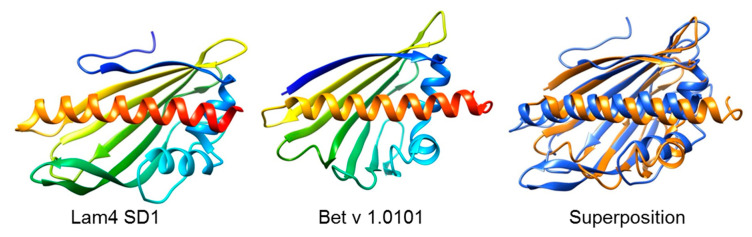
Superposition of the ribbon structure of Bet v 1.0101 (PDB 4A88, aa2-160) in orange and the structure of the StARkin domain 1 of the yeast Lam4 protein (PDB 5YQJ, aa749-929). The 3D images were created with the molecular modeling system UCSF ChimeraX (https://www.rbvi.ucsf.edu/chimerax/, accessed on 19 June 2023).

**Table 1 biomolecules-13-01151-t001:** Bet v 1-homologous pollen allergens of trees of the order Fagales.

Plant Family	Subfamily	Allergen Source	Allergen	References	UniProt/PDB
Betulaceae	Betuloideae	Birch (*Betula pendula*)	Bet v 1.0101	[6]	P15494/4A88
		Alder (*Alnus glutinosa*)	Aln g 1.0101	[8]	P38948
	Coryloideae	Hazel (*Corylus avellana*)	Cor a 1.0101	[9]	Q09407
		Hornbeam (*Carpinus betulus*)	Car b 1.0101	[19]	P38949
		Hop-hornbeam (*Ostrya carpinifolia*)	Ost c 1.0101	[18]	E2GL17
Fagaceae	Fagoideae	Beech (*Fagus silvatica*)	Fag s 1.0101	[18]	B7TWE6/6ALK
	Quercoideae	Oak (*Quercus alba*)	Que a 1.0201	[20]	B6RQS1
		Sawtooth oak *(Quercus acutissima*)	Que ac 1.0101	[21]	GenBank QOL10866.1
		Holly oak *(Quercus ilex)*	Que i 1.0101	[22]	A0A7D0TA82
		Mongolian oak (*Quercus mongolica)*	Que m 1.0101	[23]	GenBank AUH28179
	Castanoideae	Chestnut (*Castanea sativa*)	Cas s 1.0101	[18]	B7TWE3

**Table 2 biomolecules-13-01151-t002:** Plant food allergenic PR-10-like proteins.

Plant Family	Allergen Source	Allergen	References	UniProt/PDB
Actinidiaceae	Golden kiwifruit (*Actinidia chinensis*)	Act c 8.0101	[25]	D1YSM4
	Green kiwifruit (*Actinidia deliciosa*)	Act d 8.0101 Act d 11.0101	[25] [26]	D1YSM5 P85524/4IGV
Anacardiaceae	Mango (*Mangifera indica*)	Man i 2.0101	[27]	GenBank UYO79702.1
Apiaceae	Celery (*Apium graveolens*)	Api g 1.0101	[11]	P49372/2BK0
	Carrot (*Daucus carota*)	Dau c 1.0103	[28]	O04298/2WQL
Cannabaceae	Indian hemp (*Cannabis sativa*)	Can s 5.0101	[29]	I6XT51
Corylaceae	Hazelnut (*Corylus avellana*)	Cor a 1.0401	[30]	Q9SWR4/6GQ9, 6Y3H
Fabaceae	Peanut (*Arachis hypogaea*)	Ara h 8.0101	[31]	Q6VT83/4M9B
	Soybean (*Glycine max*)	Gly m 4.0101	[32]	P26987/2K7H
	Mung bean (*Vigna radiata*)	Vig r 1.0101 Vig r 6.0101	[33] [34]	Q2VU97 A0A1S3THR8/2FLH, 3C0V
Juglandaceae	English walnut (*Juglans regia*)	Jug r 5.0101	[35]	GenBank Acc. No. KX034087.1
Rosaceae	Strawberry (*Fragaria x ananassa*)	Fra a 1.0101	[36]	Q5ULZ4/6ST8
	Apple (*Malus domestica*)	Mal d 1.0101	[10]	P43211/5MMU
	Apricot (*Prunus armeniaca*)	Pru ar 1.0101	unpublished	O50001
	Cherry (*Prunus avium*)	Pru av 1.0101	[37]	O24248/1E09, 1H20
	Almond (*Prunus dulcis*)	Pru du 1.0101	[38]	B6CQS9
	Peach (*Prunus persica*)	Pru p 1.0101	[39]	Q2I6V8/6Z98
	Pear (*Pyrus communis*)	Pyr c 1.0101	[40]	O65200
	Raspberry (*Rubus idaeus*)	Rub 1 1.0101	[41]	Q0Z8U9
Solanaceae	Tomato (*Solanum lycopersicum*)	Sola l 4.0101	[42]	K4CWC5

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
