# Peer review of "The History and Science of the Major Birch Pollen Allergen Bet v 1"

_biomolecules, 2023, doi:10.3390/biom13071151_

Round 1

Reviewer 1 Report

Dear Authors, excellent work over the decades.  And scientifically interesting.

I feel some of the manuscript is a bit misleading in that homologous proteins are not all clear allergens.  There is also a curious statment on line 194. I beleive Cor a 1.0401 is a hazelnut allergen, not a pollen allergen.  I am not sure about the ligand.  Can you please check?

Do the ligands impact allergenicity, either sensitization or elicitation? And the strawberry proteins and ligands represent a complex story. 

THere seems to be disparity in allergy associated with some Bet v 1 homologous, in different sources. Is that due to route of exposure?  Or dose of protein?  For some, the relevance of allergy is weak, in some foods. 

The work of Kraft and Breiteneder has certainly helped advance the fields of molecular biology and protein biochemistry as well as allergy. Since proteins have evolved for functional properties, and not for allergy, it is a curious parallel.  Interesting too is the protein differences in different plant tissues. Many Bet v 1 genes or at least the RNA sequences differ in leaves, roots, stems and fruits. Their IgE binding potency, and allergenicity are not tested in comparative tests. Any speculation on the evolutionary differences?  Coincidences or because they have different ligands and functions?

The English language is appropriate.

Reviewer 1 Report

Comments and Suggestions for Authors

Comment 1. Dear Authors, excellent work over the decades.  And scientifically interesting.

Response 1. We thank Reviewer 1 for having taken the valuable time to contribute to the reviewing process of our article and for this positive comment.

Comment 2. I feel some of the manuscript is a bit misleading in that homologous proteins are not all clear allergens.

Response 2. We have added a paragraph to topic 4, lines 241-251, to emphasize that most of the allergens of the Bet v 1 family are found in the PR-10 subfamily. We have also added explanations why this is the case under topic 3, lines 182-188.

Comment 3. There is also a curious statement on line 194. I believe Cor a 1.0401 is a hazelnut allergen, not a pollen allergen. I am not sure about the ligand. Can you please check?

Response 3. We thank the reviewer for pointing out this mistake. Cor a 1.0401 is only present in the hazel nut. The text has been corrected, see lines 200-206.

Comment 4. Do the ligands impact allergenicity, either sensitization or elicitation?

Response 4. Indeed the ligands impact sensitization. Text has been added to illustrate this point, see lines 182-188. The case has been made especially for the ligand phytoprostane E2, see 189-198, Soh et al. 2019).

Regarding elicitation, analyses of human IgE binding to Bet v 1 in mediator release assays revealed that ligand-bound allergen induced degranulation at lower concentrations. However, in basophil activation tests with human basophils ligand-binding did not show this effect (Asam et al. 2014). This text was added, see lines 182-188.

Comment 5. And the strawberry proteins and ligands represent a complex story.

Response 5. Indeed. In terms of modulating allergenicity, ligands in general seem to paly a role. Whereas in the plant, the PR-10 proteins seem to be involved in a great variety of processes.

Comment 6. There seems to be disparity in allergy associated with some Bet v 1 homologous, in different sources. Is that due to route of exposure?  Or dose of protein?  For some, the relevance of allergy is weak, in some foods.

Response 6. The ability to mediate sensitization and to elicit allergic symptoms is quite low among the Bet v 1 homologs from plant foods. This depends on the route of exposure (inhalation versus ingestion (most Bet v 1 homologues are not resistant to proteolysis), the dose of the protein (Bet v 1 is a very prominent protein in pollen), and the type of ligand bound. We have added a sentence and a reference, see lines 142-143.

Comment 7. The work of Kraft and Breiteneder has certainly helped advance the fields of molecular biology and protein biochemistry as well as allergy. Since proteins have evolved for functional properties, and not for allergy, it is a curious parallel.

Response 7. Exactly, these proteins have evolved for broad range of very specific functions and the triggering of allergy is to be considered an “undesirable” by-product.

Comment 8. Interesting too is the protein differences in different plant tissues. Many Bet v 1 genes or at least the RNA sequences differ in leaves, roots, stems and fruits. Their IgE binding potency, and allergenicity are not tested in comparative tests. Any speculation on the evolutionary differences?  Coincidences or because they have different ligands and functions?

Response 8. Bet v 1 homologs from various tissues of birch have not been tested for their capacity to bind IgE or in BAT assays. The Bet v 1 scaffold is a very old and versatile one and has been used for a wide variety of functions. We do not believe that the Bet v 1 homologs present in birch or in other organisms are the result of convergent evolution. Rather it is a very useful scaffold that was conserved and put to many uses. The Bet v 1-like superfamily, or as it is also called, the StARkin superfamily, keeps growing and new members are still being discovered.

Reviewer 2 Report

In the present review H. Breiteneder and D. Kraft perfectly recapitulate the history of science in regard of the major birch pollen allergen Bet v 1, which was the first cloned and recombinantly expressed allergenic protein.

The review, in a very enjoyable way, summarizes all the background how the idea of establishing recombinant allergens arise and was implemented. Furthermore, all important key features of Bet v 1 and the PR-10 family in general, as well as the resulting conclusions having a high impact on the understanding of allergenic proteins are depicted and described.

Therefore, only some minor comments can be addressed:

- a general sentence summarizing the physicochemical parameters of Bet v 1, or in general the PR-10 family, would be nice to have, e.g. low stability, low melting point (compared for example to PR-14 family, high stability (proteolytic and thermical) higher melting temperature), pI, size

- in the last paragraph Bet v 1-related non allergenic molecules from bacteria are described. Here one similar molecular to my point of view is missing, describing a Bet v 1 structurally homologous molecule from meadow rue (as well a plant, so more related to Bet v 1?) using as toolbox for epitope definition (Berkner H. et al., DOI: 10.1371/journal.pone.0111691).

- maybe a short conclusion and/or summary at the end of the manuscript would be nice (if the journal guidelines allow this)

- is reference 6 really the correct citation of Api g 1? Should be DOI: 10.1111/j.1432-1033.1995.484_2.x

- line 136, check sentence: reacting to n foods, maybe the n should be deleted?

Reviewer 1 Report

Comments and Suggestions for Authors

In the present review H. Breiteneder and D. Kraft perfectly recapitulate the history of science in regard of the major birch pollen allergen Bet v 1, which was the first cloned and recombinantly expressed allergenic protein.

The review, in a very enjoyable way, summarizes all the background how the idea of establishing recombinant allergens arise and was implemented. Furthermore, all important key features of Bet v 1 and the PR-10 family in general, as well as the resulting conclusions having a high impact on the understanding of allergenic proteins are depicted and described.

Therefore, only some minor comments can be addressed:

Response. We thank the reviewer for having taken the time to examine our manuscript and are grateful for the positive feedback and the comments.

Comment 1. A general sentence summarizing the physicochemical parameters of Bet v 1, or in general the PR-10 family, would be nice to have, e.g. low stability, low melting point (compared for example to PR-14 family, high stability (proteolytic and thermical) higher melting temperature), pI, size.

Response 1. We thank the reviewer for this suggestion. A corresponding description has been added in lines 112-117.

Comment 2. - in the last paragraph Bet v 1-related non allergenic molecules from bacteria are described. Here one similar molecular to my point of view is missing, describing a Bet v 1 structurally homologous molecule from meadow rue (as well a plant, so more related to Bet v 1?) using as toolbox for epitope definition (Berkner H. et al., DOI: 10.1371/journal.pone.0111691).

Response 2. We thank the reviewer for this suggestion. We have added the corresponding text in lines 265-268 .

Comment 3. - maybe a short conclusion and/or summary at the end of the manuscript would be nice (if the journal guidelines allow this).

Response 3. We have added a short conclusion at the end of the article. See lines 319-330.

Comment 4. - is reference 6 really the correct citation of Api g 1? Should be DOI: 10.1111/j.1432-1033.1995.484_2.x

Response 4. We apologize for this mistake. The correct reference was added. See line 100.

Comment 5. - line 136, check sentence: reacting to n foods, maybe the n should be deleted?

Response 5. Sorry, we did not find this phrase “reacting to n foods”.